# Patient Characteristics, Treatment Patterns, and Outcomes in Unresectable Hepatocellular Carcinoma Treated with First-Line Systemic Therapy in the United States

**DOI:** 10.3390/cancers17213499

**Published:** 2025-10-30

**Authors:** Nguyen H. Tran, Scott A. Soefje, Nivedita Rangarajan, Purushotham Krishnappa, Tyler E. Wagner, Stephen J. Valerio, Rye Anderson, Jody C. Olson

**Affiliations:** 1Department of Oncology, Mayo Clinic, Rochester, MN 55905, USA; 2Department of Pharmacy, Mayo Clinic, Rochester, MN 55905, USA; 3nference, inc., Cambridge, MA 02142, USApurushotham.k@nference.net (P.K.);; 4US Medical Affairs, AstraZeneca, Gaithersburg, MD 20878, USA; 5US Oncology Care and Access, AstraZeneca, Gaithersburg, MD 20878, USA; rye.anderson@astrazeneca.com; 6Division of Gastroenterology and Hepatology, Mayo Clinic, Rochester, MN 55905, USA

**Keywords:** gastrointestinal bleeding, immunotherapy, systemic therapy, treatment patterns, unresectable hepatocellular carcinoma, real-world evidence

## Abstract

**Simple Summary:**

This study offers an alternative framework for the assessment of gastrointestinal (GI) bleeding risk in patients with unresectable hepatocellular carcinoma (uHCC). Approximately two-thirds of the patients in this study had GI bleeding risk, and overall survival rates were lower in patients with versus without GI bleeding risk, highlighting the complexity of uHCC and the unmet need for guidance on characteristics-driven treatment decisions.

**Abstract:**

**Background:** Immunotherapy-based regimens have expanded the treatment landscape for unresectable hepatocellular carcinoma (uHCC); however, real-world data are limited. **Methods:** This retrospective, observational study used data from electronic medical records from Mayo Clinic sites across the United States. Patients with uHCC who initiated a first-line (1L) systemic therapy between June 2020–October 2022 with ≥2 follow-up visits were included. Treatment patterns, overall survival (OS), and post-index gastrointestinal (GI) bleeding were assessed by GI bleeding risk defined by Child–Pugh Class B or C, pre-index GI bleeding, uncontrolled hypertension, or significant varices and band ligation. **Results:** Of 186 included patients, 68.8% had GI bleeding risk and 31.2% did not. Atezolizumab plus bevacizumab was the most common 1L systemic therapy in patients with or without GI bleeding risk (72.7% and 29.3%, respectively). Median OS (95% confidence interval) with atezolizumab plus bevacizumab was 12.8 (8.0–19.3) months and not reached in patients with and without GI bleeding risk, respectively. OS rates with atezolizumab plus bevacizumab in patients with or without GI bleeding risk, respectively, were 52.3% and 70.6% at 12 months, 41.6% and 57.8% at 18 months, and 34.6% and 51.3% at 24 months. Post-index GI bleeding with atezolizumab plus bevacizumab occurred in 19.4% and 5.9% of patients with and without GI bleeding risk, respectively. **Conclusions:** During this study period, atezolizumab + bevacizumab was the most common 1L therapy for patients with uHCC, regardless of GI bleeding risk. OS rates with atezolizumab + bevacizumab were lower in patients with versus without GI bleeding risk. Findings highlight the unmet need for guidance on characteristics-driven treatment decisions.

## 1. Introduction

Liver cancer is the sixth most common cancer and has the third highest mortality rate of cancers worldwide. Hepatocellular carcinoma (HCC) is the most common form of primary liver cancer [1]. Unfortunately, many people with HCC are diagnosed at an advanced stage, where curative-intent therapy is not an option [2].

It was estimated that there would be 42,240 new cases relating to liver and intrahepatic bile duct cancer in the United States (US), with an estimated 30,090 deaths from such cancers in 2025 [3]. Over the past three decades, the incidence of HCC in the US has been increasing and is projected to continue through to 2030 [4].

First-line (1L) systemic therapy for unresectable HCC in the US was previously limited to tyrosine kinase inhibitors, sorafenib, and lenvatinib [5]. However, the treatment landscape for patients with unresectable HCC (uHCC) in the US has expanded with the introduction of immunotherapy-based regimens, with the approval of atezolizumab plus bevacizumab (an anti-programmed cell death ligand-1 [anti-PD-L1] and anti-vascular endothelial growth factor regimen) in May 2020 based on the results of the IMbrave150 study (NCT03434379) [6,7] and the STRIDE regimen (Single Tremelimumab Regular Interval Durvalumab; anti-cytotoxic T lymphocyte-associated antigen 4 and anti-PD-L1) in October 2022 based on the results from the HIMALAYA study (NCT03298451) [8,9]. In April 2025, the US Food and Drug Administration approved the use of nivolumab plus ipilimumab in patients with uHCC based on the CheckMate 9DW study [10], which demonstrated a statistically significant overall survival (OS) benefit versus lenvatinib or sorafenib [11,12]. Camrelizumab plus rivoceranib has also shown a statistically significant benefit in progression-free survival (PFS) and OS compared with sorafenib in patients with uHCC in the CARES-310 study [13]. However, there are other immunotherapy-based regimens that have demonstrated limited clinical improvement in advanced HCC, including lenvatinib plus pembrolizumab, which did not significantly improve OS and PFS versus lenvatinib plus placebo in the LEAP-002 study [14], and atezolizumab plus cabozantinib, which demonstrated statistically significant improvement versus sorafenib in PFS but not OS in the COSMIC-312 study [15].

In addition to tumor burden, treatment of HCC is further complicated by the underlying liver disease [16]. The majority of patients with HCC have cirrhosis, which can lead to portal hypertension and esophageal varices, putting them at increased risk for gastrointestinal (GI) bleeding, independent of treatment [17]. Upon the approval of atezolizumab plus bevacizumab, the balance of bleeding risk versus clinical benefit was an important consideration that drove real-world treatment decision-making. In a systematic review and meta-analysis of patients with HCC, higher risk of upper GI bleeding was observed in those who received atezolizumab and bevacizumab, compared with those who received tyrosine kinase inhibitors [18]. Endoscopic evaluation and management of varices is recommended prior to the initiation of atezolizumab plus bevacizumab [5]. However, a standardized definition to identify bleeding risk in uHCC has not yet been established [19], and there is a growing need to inform treatment decision-making.

This study aimed to describe patient characteristics, treatment patterns, and outcomes in patients with uHCC with or without GI bleeding risk treated with 1L systemic therapy in the US after the approval of atezolizumab plus bevacizumab and prior to the approval of other immunotherapies for uHCC.

## 2. Materials and Methods

### 2.1. Data Source

This retrospective, observational study used data from adult patients with uHCC from electronic medical records from Mayo Clinic sites in the US.

### 2.2. Study Population

Patients with uHCC (18 to 90 years of age) from the Mayo Clinic sites who received a 1L systemic therapy between June 2020 and October 2022, with at least two follow-up visits, were included in the analysis. HCC diagnosis was captured using International Classification of Diseases criteria 9th edition (ICD-9) codes 155.0 and 155.2, 10th edition (ICD-10) codes C22.0, C22.8, and C22.9, or using natural language processing models. Unresectability was based on the surgeon’s assessment of specific criteria, including status of infiltrative disease, Barcelona Clinic Liver Cancer (BCLC) C staging, and comorbidities limiting resection. Patients were required to have at least two follow-up visits to observe continuity of care and to allow assessment of post-index events.

The index date was defined as date of first administration of 1L systemic therapy and was restricted to between June 2020 and October 2022, which was the time period between the approval of atezolizumab plus bevacizumab and prior to the approval of other immunotherapies for uHCC in the US.

### 2.3. Study Variables

Patient characteristics, including maximum line of therapy, Child–Pugh Class, albumin–bilirubin (ALBI) grade, HCC etiology, comorbidities or risk factors, any locoregional therapy (LRT), pre-index GI bleeding, and the number of patients who underwent esophagogastroduodenoscopy (EGD) were assessed in the full cohort.

Child–Pugh score was identified using a Regex search for mentions of Child–Pugh score or Class in notes and by calculation of score based on components. Child–Pugh scores were calculated based on the following components that were present within the 6-month pre-index period: total bilirubin, albumin, international normalized ratio, ascites severity, and encephalopathy grade [20]. For ascites severity and encephalopathy grade, the presence of an ICD code or a positive mention in the notes was assigned a score of 2; lack of ICD code or no mention in the notes was assigned a score of 1. A missing component was assigned a score of 1. Child–Pugh scores were estimated only for patients with 3 or more components present.

ALBI grade was identified using a Regex search for mentions of ALBI score or grade in notes and by calculation of grade based on components. ALBI scores were calculated using albumin (g/L) and bilirubin (µmol/L) levels using the following formula: (log_10_ bilirubin × 0.66) + (albumin × −0.085).

When the information was not available in the electronic medical records, BCLC stage was calculated in patients with at least three components present out of Eastern Cooperative Oncology Group performance status, Child–Pugh Class, tumor number, tumor size, and portal vein invasion. In patients with two or fewer components present, BCLC stage was classified as unknown.

Etiologies reported included autoimmune hepatitis, hepatitis B virus, hepatitis C virus (HCV), hemochromatosis, and metabolic dysfunction-associated steatohepatitis (MASH), which included nonalcoholic steatohepatitis and fatty liver disease to align with the updates to the nomenclature for fatty liver disease [21]. Comorbidities or risk factors reported included hypertension, alcoholism, cirrhosis, diabetes, obesity, and smoking. LRTs captured included ablation, transarterial chemoembolization, external beam radiation therapy, percutaneous ethanol injection, stereotactic body radiation therapy, and transarterial radioembolization. GI bleeding at any time point and within 6 months prior to index were assessed. The number of patients who underwent EGD was assessed, and among those, the number of patients with varices was reported; variceal grading was presented if these data were available.

In the absence of a standardized definition, risk of GI bleeding was defined as meeting at least one of the following criteria: patients with Child–Pugh Class B or C, patients with GI bleeding during the 6-month pre-index period, patients with uncontrolled hypertension (defined as having at least two classes of anti-hypertensive medications or at least two vital entries of systolic blood pressure > 140 mmHg or diastolic blood pressure > 90 mmHg), or patients who underwent EGD procedure and had significant varices (classified as Grade 2 or 3 or large varices) and band ligation.

Treatment patterns, including type of 1L and second-line (2L) systemic therapies received, were assessed by risk of GI bleeding. Outcomes, including OS and post-index GI bleeding, were evaluated in the full cohort and by risk of GI bleeding. Patients were censored according to their last recorded Mayo Clinic visit date that occurred within the study period if they had not died by January 2024 (the end of the study period). Outcomes were also assessed in a subgroup of patients who received atezolizumab plus bevacizumab, which included those who received bevacizumab within 30 days from initiation of atezolizumab.

### 2.4. Statistical Analyses

Categorical variables were described using frequency counts or percentages for each category. An exploratory analysis of OS was conducted using the Kaplan–Meier method to estimate survival probabilities. *p*-Values for this exploratory analysis were derived from the log-rank test, comparing survival between patients with versus without GI bleeding risk.

## 3. Results

### 3.1. Patient Characteristics

A total of 186 patients met the inclusion criteria (Figure 1). The median age of this patient cohort was 68 years. Clinical characteristics are presented in Table 1 for patients treated with 1L systemic therapy.

MASH (42.7%) and HCV (40.2%) were the most common etiologies of HCC reported. Hypertension (94.6%), smoking (80.6%), and cirrhosis (74.2%) were the most common comorbidities in the full cohort. There were 62/117 (53.0%) patients with Child–Pugh Class B and 50/117 (42.7%) patients with Child–Pugh Class A. In this cohort, 65/173 (37.6%) patients had ALBI Grade 1, 94/173 (54.3%) had ALBI Grade 2, and 14/173 (8.1%) had ALBI Grade 3. There were 23/186 (12.4%) patients with BCLC Stage A, 66/186 (35.5%) with BCLC Stage B, 62/186 (33.3%) with BCLC Stage C, and 4/186 (2.2%) with BCLC Stage D. The BCLC stage was unknown in 31/186 (16.7%) patients.

Pre-index GI bleeding was reported in 41/186 (22.0%) patients, and of these cases, 25/41 (61.0%) occurred within 6 months pre-index. A total of 116/186 (62.4%) of patients underwent EGD, of which 84/116 (72.4%) had varices of any grade.

In this cohort, 128/186 (68.8%) patients had GI bleeding risk and 58/186 (31.2%) did not have GI bleeding risk (Table 2). The most common reason that patients were considered to have GI bleeding risk was uncontrolled hypertension, 98/128 (76.6%).

### 3.2. Treatment Patterns

Atezolizumab plus bevacizumab (72.7%) and atezolizumab monotherapy (9.4%) were the most common 1L systemic therapies received overall for patients with GI bleeding risk during the timeframe of this analysis (Figure 2A). The most common 1L systemic therapies for patients without GI bleeding risk were atezolizumab plus bevacizumab (29.3%), nivolumab plus ipilimumab (13.8%), and bevacizumab plus chemotherapy (12.1%; Figure 2A).

For patients with risk of GI bleeding, 112/128 (87.5%) patients received a 1L therapy that contained atezolizumab, in which 93 patients received atezolizumab plus bevacizumab, 12 received atezolizumab monotherapy, and 7 received other atezolizumab-containing combination therapies. For patients without risk of GI bleeding, 22/58 (37.9%) patients received a 1L therapy containing atezolizumab, in which 17 patients received atezolizumab plus bevacizumab and five patients received atezolizumab monotherapy. In total, 16/58 (27.6%) patients without GI bleeding risk and 31/128 (24.2%) patients with GI bleeding risk received a 2L systemic therapy, of which lenvatinib was the most common 2L systemic therapy for patients in both groups (Figure 2B).

### 3.3. Overall Survival

Median follow-up (interquartile range) was 20.6 (12.9–26.9) months in all patients, 20.9 (12.3–25.8) months in patients with GI bleeding risk, and 20.4 (14.3–28.2) months in patients without GI bleeding risk (Figure 3). Median OS (95% confidence interval [CI]) was 14.4 (10.1–19.4) months in all patients (Figure 3A), and was shorter in patients with GI bleeding risk (11.4 [7.8–14.8] months) versus patients without GI bleeding risk (40.3 [not calculable] months) (Figure 3B). The OS rates for patients treated with any systemic therapy with and without GI bleeding risk were 48.9% and 68.9% at 12 months, 38.1% and 61.1% at 18 months, and 31.8% and 56.8% at 24 months, respectively (Figure 3B). Median OS for patients with pre-index GI bleeding risk (n = 41) and patients with pre-index GI bleeding risk within 6 months of the pre-index date (n = 25) was 12.5 months (Figure 4).

For patients treated with atezolizumab plus bevacizumab, in the full cohort, median follow-up (interquartile range) was 21.3 (13.4–27.0) months in all patients, 21.6 (13.1–27.0) months in patients with GI bleeding risk, and 20.3 (15.9–27.2) months in patients without GI bleeding risk (Figure 5). In those patients, median OS (95% CI) was 13.9 (8.7–19.4) months in all patients (Figure 5A), 12.8 (8.0–19.3) months in patients with GI bleeding risk, and not reached in patients without GI bleeding risk (Figure 5B). The OS rates for patients treated with atezolizumab plus bevacizumab, with and without GI bleeding risk, were 52.3% and 70.6% at 12 months, 41.6% and 57.8% at 18 months, and 34.6% and 51.3% at 24 months, respectively (Figure 5B).

### 3.4. Post-Index GI Bleeding

Overall, post-index GI bleeding was observed in 32/186 (17.2%) patients, including 30/128 (23.4%) patients with GI bleeding risk and 2/58 (3.4%) patients without GI bleeding risk. Among patients who received atezolizumab plus bevacizumab, post-index GI bleeding was observed in 19/110 (17.3%) patients, including 18/93 (19.4%) patients with GI bleeding risk and 1/17 (5.9%) patient without GI bleeding risk.

## 4. Discussion

This retrospective study evaluated patient characteristics, treatment patterns, and outcomes in patients with uHCC treated with 1L systemic therapy in the US after the approval of atezolizumab plus bevacizumab and prior to the approval of other immunotherapy regimens for uHCC. Patients with HCC have an increased risk of GI bleeding due to their underlying cirrhosis [19]. Real-world studies are important for providing clinically informative data; however, there is a lack of a standardized definition for bleeding risk that could be used in the clinical setting. In this study, we described a method of identifying patients at increased risk of bleeding using available clinical information, which could be beneficial not only for treatment decisions but also for future real-world studies in HCC.

The most common comorbidities or risk factors observed in this cohort were smoking, hypertension, cirrhosis, and diabetes, which are known risk factors for HCC [5,22,23,24]. Compared with other real-world studies in patients with HCC who received systemic therapy in the US, the proportion of patients with history of smoking, or who had hypertension, diabetes, or cirrhosis, was higher in our study [22,25]. The most common HCC etiologies reported in the full cohort were MASH and HCV, which are the prevalent etiologies in the US [5,26]. In this study, more than 50% of patients had a Child–Pugh Class B uHCC or had ALBI Grade 2 or 3. Phase 3 studies in this setting, including the IMbrave150 study [6], only include patients with Child–Pugh Class A uHCC, and in the case of the IMbrave150 study, most patients had ALBI Grade 1 [27], suggesting that the patient population in our current study had worse liver function, which is generally consistent with another real-world study in the US [22]. In general, patients with worse liver function have a decreased life expectancy [28], and ALBI grade has been reported to be a prognostic factor for outcomes with atezolizumab plus bevacizumab in patients with HCC [27]. Additionally, diabetes and insulin resistance have been shown to have a negative impact on HCC prognosis, both of which are associated with an increased risk of metastases and shorter survival times compared with patients with HCC without these comorbidities [24]. Older age has also been associated with high mortality in patients with diabetes and HCC. Insulin-resistant drugs, such as metformin, have been shown to improve HCC prognosis and reduce mortality, but more studies are needed to determine appropriate treatment guidelines in this patient population [24].

In this study, risk of GI bleeding was defined as having uncontrolled hypertension, Child–Pugh Class B or C, having experienced a GI bleeding event, or having undergone EGD procedure and had significant varices (classified as Grade 2 or 3 or large) and band ligation. Approximately two-thirds of patients in this study were at risk of GI bleeding by this definition, suggesting that patients treated in clinical practice carried increased risk for GI bleeding compared to clinical study populations. Notably, 17.2% of patients in the current analysis had post-index GI bleeding, while in the IMbrave150 study, upper GI bleeding was observed in 7.0% of patients in the atezolizumab plus bevacizumab group and in 4.5% of patients in the sorafenib group, reflecting the rigorous EGD screening protocol and exclusion of patients with untreated or incompletely treated varices in the IMbrave150 study [6].

From June 2020 to October 2022, atezolizumab plus bevacizumab was reported to be the most commonly used 1L therapy in our study for both patients with and without GI bleeding risk. Previously, bevacizumab has been reported to alter vascular integrity and has been associated with GI bleeding risk [18]. However, in this study, a higher proportion of patients received atezolizumab plus bevacizumab in the GI bleeding risk cohort versus the no GI bleeding risk cohort. This observation may be due to the absence of clinical differentiation in most patients at the time of analysis due to limited availability of other approved immunotherapy regimens. In addition, the 1L treatment profile for patients without GI bleeding risk was more diverse compared with patients with GI bleeding risk, suggesting that healthcare provider preference may play a role in treatment selection. Healthcare providers might not have been comfortable with the atezolizumab plus bevacizumab combination when it was initially approved and favored other immune checkpoint therapies, or perceived these patients to be ineligible to receive this combination treatment. Data for 2L therapies in both groups were sparse (Figure 2); most patients received 2L lenvatinib regardless of GI bleeding risk. A previous study by our team found that in patients with advanced HCC, there were no significant differences in survival or time to treatment discontinuation between immune checkpoint inhibitors or anti-vascular endothelial growth factor therapy following 1L atezolizumab + bevacizumab [29]. It should be acknowledged that uncontrolled hypertension is not commonly considered as a risk factor for upper GI bleeding in patients with cirrhosis; therefore, interpretation of risk of bleeding may vary across healthcare providers. However, the sample sizes for these cohorts are small, and results should be interpreted with caution.

Off-label use of atezolizumab was also observed in our current study, including the use of atezolizumab monotherapy. Possible reasons for this approach might have been to avoid the contraindications of bevacizumab in some patients or as a form of therapy until EGD could be performed to further assess GI bleeding risk, though the small number of patients precludes further analysis of this group. In the HIMALAYA study, no EGD was required for inclusion in the study, and no treatment-related GI or esophageal bleeding was reported with STRIDE [8]; however, this regimen was not approved for uHCC during this study period. The lack of bleeding events with STRIDE in the HIMALAYA study may partly explain the longer survival benefit with STRIDE versus sorafenib at 5 years [30]. It should be acknowledged that data for patients who received atezolizumab plus bevacizumab in the IMbrave150 study were available up to a median follow-up of 17.6 months [31], while for patients who received STRIDE in the HIMALAYA study, data were available up to a median follow-up of 62.49 months [30].

In our exploratory analysis of survival outcomes, median OS with atezolizumab and bevacizumab was 12.8 months in patients with GI bleeding risk (N = 93) and not reached in patients without GI bleeding risk (N = 17), which may be due to the small sample size. At 12, 18, and 24 months, OS rates with atezolizumab and bevacizumab were lower in patients with versus without GI bleeding risk. Furthermore, patients with GI bleeding risk had a higher rate of post-index GI bleeding versus those without GI bleeding risk. Out of 19 patients receiving atezolizumab plus bevacizumab who had post-index GI bleed, 18 had risk of GI bleeding. These findings suggested that, in the absence of a standardized definition, the approach to cohort selection in this study was successful in identifying patients at risk of GI bleeding. In a Canadian real-world study investigating the outcomes of atezolizumab plus bevacizumab in patients with HCC [32], the median OS was 20.3 months (95% CI, 13.0 months–not reached) in patients who experienced bleeding and 19.7 months (95% CI, 16.5 months–not reached) in patients who did not experience bleeding. Differences in OS between studies may be due to variations in study cohort eligibility, geographic region, and healthcare systems. In a recent systematic review and meta-analysis of risk of bleeding in patients with HCC receiving atezolizumab plus bevacizumab therapy, there was an increased risk of GI bleeding in comparison with tyrosine kinase inhibitors, and higher ALBI grade and high body mass index were found to predict bleeding complications [18]. Furthermore, a real-world observational study reported that hepatic decompensation was identified with worse prognosis in patients receiving atezolizumab plus bevacizumab, and was found to be more common in patients with nonviral etiology and ALBI grade greater than 1 at baseline [33]. Child–Pugh Class B and C, which indicated worsening liver function, were included as one of the criteria for bleeding risk in our current study. Overall, these findings indicate that the benefit of atezolizumab plus bevacizumab may be limited in patients with GI bleeding risk, and further real-world studies are warranted to inform optimal patient selection in clinical practice.

The limitations in this analysis are common to those of a retrospective, observational study, and therefore, it was subject to possible selection bias. To note, there may be survival bias as participants needed to have at least 2 visits to be included in the study. It is also relevant to note that the findings of this study relied on the completeness and accuracy of electronic medical record entries, which may have contributed to the small sample sizes. This study was not powered to make formal statistical comparisons between groups.

The data reported in the current study were collected from Mayo Clinic sites across the US; therefore, the generalizability to other providers or other countries may be limited. Furthermore, the patients included in this analysis were predominantly white and, therefore, even within the US population, results may not be generalizable as they are not representative of the entire US population.

## 5. Conclusions

In the US, prior to the approval of other 1L systemic therapies, many patients received atezolizumab plus bevacizumab as 1L systemic therapies, despite the risk of GI bleeding. Assessment of survival outcomes demonstrated that patients with GI bleeding risk had lower OS rates with atezolizumab plus bevacizumab than those without GI bleeding risk. When choosing treatment for patients with uHCC, risk of GI bleeding should be considered; however, currently, there is no universal definition for assessment of this risk in this clinical context. This study offers an alternative framework for the assessment of GI bleeding risk in patients with uHCC, which could be used in future real-world studies and might assist treatment selection. Taken together, these findings highlight the complexity of uHCC and highlight the unmet need for guidance on characteristics-driven treatment decisions.

## Figures and Tables

**Figure 1 cancers-17-03499-f001:**
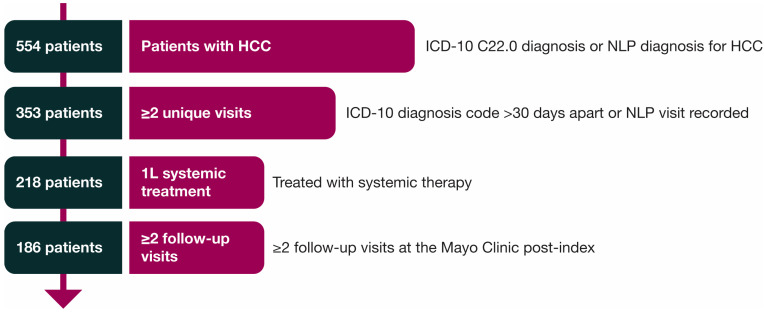
Selection of study cohort. 1L = first-line; HCC = hepatocellular carcinoma; ICD = International Classification of Diseases; NLP = natural language processing.

**Figure 2 cancers-17-03499-f002:**
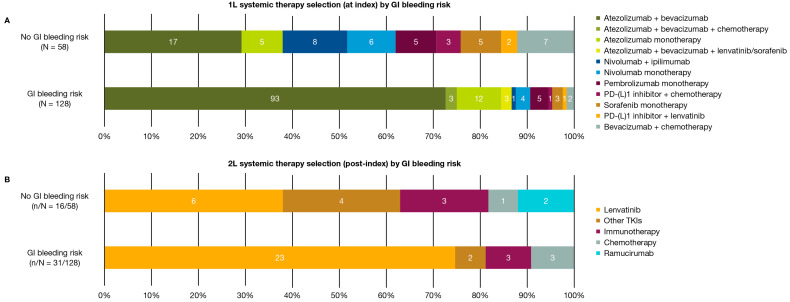
1L (**A**) and 2L (**B**) treatment patterns in all patients by GI bleeding risk. Chemotherapy included one or more of the following: cisplatin, doxorubicin, fluorouracil, irinotecan, leucovorin, oxaliplatin, or temozolomide; PD-(L)1 inhibitors included atezolizumab, nivolumab, and pembrolizumab; other TKIs included cabozantinib, regorafenib, or sorafenib, and one patient in the no GI bleeding risk group who received cabozantinib + ipilimumab + nivolumab; immunotherapy included ipilimumab + nivolumab, or pembrolizumab. 1L = first-line; 2L = second-line; GI = gastrointestinal; N = number of patients in group; n = number of patients receiving 2L systemic therapy; PD-(L)1 = programmed cell death (ligand)-1; TKI = tyrosine kinase inhibitor.

**Figure 3 cancers-17-03499-f003:**
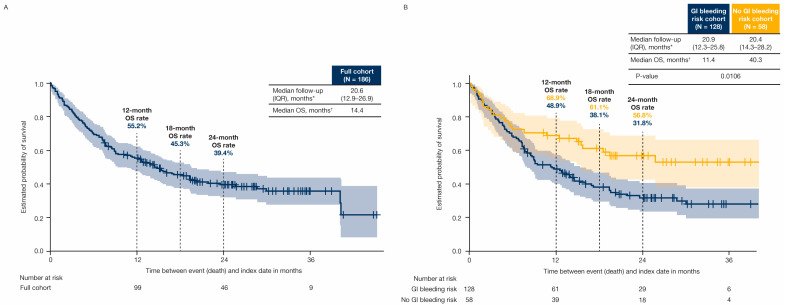
OS from index for all patients in the (**A**) full cohort and by (**B**) GI bleeding risk. Shaded regions represent 95% CI. * Median follow-up is calculated from index date to last record at US Mayo Clinic. † Patients were censored according to their last recorded Mayo Clinic visit date that occurred within the study period if they had not died by January 2024 (the end of the study period). CI = confidence interval; GI = gastrointestinal; IQR = interquartile range; N = number of patients in group; OS = overall survival.

**Figure 4 cancers-17-03499-f004:**
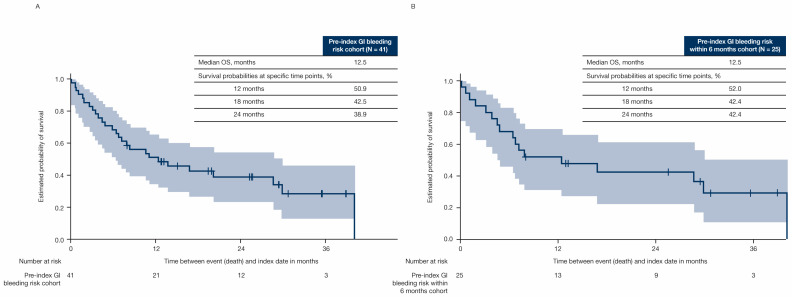
OS for patients with (**A**) pre-index GI bleeding risk and (**B**) pre-index GI bleeding risk within 6 months of the pre-index date. Shaded regions represent 95% CI. CI = confidence interval; GI = gastrointestinal; N = number of patients in group; OS = overall survival.

**Figure 5 cancers-17-03499-f005:**
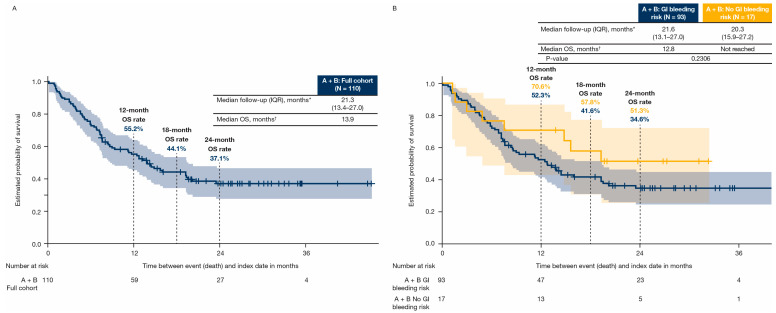
OS from index in the atezolizumab + bevacizumab cohort in the (**A**) full cohort and by (**B**) GI bleeding risk. Shaded regions represent 95% CI; a patient was included in the atezolizumab + bevacizumab cohort if bevacizumab was administered within 30 days from initiation of atezolizumab. * Median follow-up is calculated from index date to last record at US Mayo Clinic. † Patients were censored according to their last recorded Mayo Clinic visit date that occurred within the study period if they had not died by January 2024 (the end of the study period). A = atezolizumab; B = bevacizumab; CI = confidence interval; GI = gastrointestinal; IQR = interquartile range; N = number of patients in group; OS = overall survival.

**Table 1 cancers-17-03499-t001:** Full cohort demographics and disease characteristics.

	Full Cohort (N = 186)
Maximum number of line of therapy, n (%)	
1	136 (73.1)
2	36 (19.4)
3	10 (5.4)
4	4 (2.2)
Child–Pugh Class, n/N (%) *	
A	50/117 (42.7)
B	62/117 (53.0)
C	5/117 (4.3)
ALBI Grade, n/N (%)	
1	65/173 (37.6)
2	94/173 (54.3)
3	14/173 (8.1)
BCLC Stage, n/N (%)	
A	23 (12.4)
B	66 (35.5)
C	62 (33.3)
D	4 (2.2)
Unknown	31 (16.7)
Etiologies, n/N (%)	
Autoimmune hepatitis	2/82 (2.4)
HBV	19/82 (23.2)
HCV	33/82 (40.2)
Hemochromatosis	6/82 (7.3)
MASH ^†^	35/82 (42.7)
PBC	3/82 (3.7)
Comorbidities/risk factors, n (%) ^‡^	
Alcoholism	53 (28.5)
Cirrhosis	138 (74.2)
Diabetes	111 (59.7)
Hypertension	176 (94.6)
Obesity	90 (48.4)
Smoker	150 (80.6)
Any LRT, n (%)	108 (58.1)
Ablation	30 (16.1)
EBRT	30 (16.1)
TACE	64 (34.4)
PEI	1 (0.5)
SBRT	12 (6.5)
TARE	83 (44.6)
Other relevant codes	49 (26.3)
Pre-index GI bleeding, n (%)	41 (22.0)
Within 6 months pre-index, n/N (%)	25/41 (61.0)
Patients with EGD, n (%)	116 (62.4)
Patients with EGD and varices, n/N (%)	84/116 (72.4)
Patients with EGD, varices, and variceal grading, n/N (%)	36/116 (31.0)
1	19/116 (16.4)
2	15/116 (12.9)
3	2/116 (1.7)

Percentages were calculated as n/N with N = 186 unless otherwise noted. * Child–Pugh Classes were identified using a Regex search for mentions of Child–Pugh score or class in notes and/or by calculation of score based on components. ^†^ Including NASH and FLD. ^‡^ Comorbidities and risk factors are not mutually exclusive. ALBI = albumin–bilirubin; BCLC = Barcelona Clinic Liver Cancer; EBRT = external beam radiation therapy; EGD = esophagogastroduodenoscopy; FLD = fatty liver disease; GI = gastrointestinal; HBV = hepatitis B virus; HCV = hepatitis C virus; LRT = locoregional therapy; MASH = metabolic dysfunction-associated steatohepatitis; N = number of patients in group; n = number of patients with demographic/disease characteristic; NASH = nonalcoholic steatohepatitis; PBC = primary biliary cholangitis; PEI = percutaneous ethanol injection; SBRT = stereotactic body radiation therapy; TACE = transarterial chemoembolization; TARE = transarterial radioembolization.

**Table 2 cancers-17-03499-t002:** Patient profile by risk of GI bleeding.

	Full Cohort (N = 186)
Patients with risk of GI bleeding, n (%)	128 (68.8)
Uncontrolled hypertension *, n/N (%)	98/128 (76.6)
Child–Pugh Class B or C, n/N (%)	67/128 (52.3)
GI bleeding event, n/N (%)	25/128 (19.5)
Significant varices and band ligation, n/N (%)	6/128 (4.7)
Patients without risk of GI bleeding, n (%)	58 (31.2)

Risk of GI bleeding was assessed 6 months before index (1L treatment initiation). Criteria for risk of GI bleeding are not mutually exclusive. * Defined as received ≥2 classes of anti-hypertensive medications or ≥2 vital entries of systolic blood pressure > 140 mmHg or diastolic blood pressure > 90 mmHg. 1L = first-line; GI = gastrointestinal.

## Data Availability

The data that support the findings of this study are derived from the Mayo Clinic EHR, and are available under license from the Mayo Clinic and nference. The data used for this study cannot be shared publicly as the sharing of individuals’ protected health information is forbidden.

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
