# Peer review of "Patient Characteristics, Treatment Patterns, and Outcomes in Unresectable Hepatocellular Carcinoma Treated with First-Line Systemic Therapy in the United States"

_cancers, 2025, doi:10.3390/cancers17213499_

Round 1

Reviewer 1 Report

Comments and Suggestions for Authors

The topic is important although i have some concerns on the novelty of the paper.

Please remove the confidence intervals from the KM curves

THe authors should comment more on the impact of comorbidities on patient outcomes, in this regard cite the relevant review PMID: 23845075 

What is the impact of subesequent therapies on post-progression survival?

The authors should explain more in details the criteria for unresectability

Author Response

Comments 1: Please remove the confidence intervals from the KM curves.

Response 1: As requested, we have removed the 95% confidence intervals from the tables located in Figures 3 and 4. We have retained the confidence interval shading in the graphs to show the variation in values across all data points. (Page #8)

Comments 2: The authors should comment more on the impact of comorbidities on patient outcomes, in this regard cite the relevant review PMID: 23845075

Response 2: Please note we have now included the relevant review and a small section of text dedicated to highlighting the associations between diabetes, insulin resistance, and age with HCC prognosis and outcomes. The following text is now included in the Discussion: ‘Additionally, diabetes and insulin resistance have been shown to have a negative impact on HCC prognosis, both of which are associated with an increased risk of metastases and shorter survival times compared with patients with HCC without these comorbidities (24). Older age has also been associated with high mortality in patients with diabetes and HCC. Insulin-resistant drugs, such as metformin, have been shown to improve HCC prognosis and reduce mortality, but more studies are needed to determine appropriate treatment guidelines in this patient population (24).’ (Page #10)

Comments 3: What is the impact of subsequent therapies on post-progression survival?

Response 3: Thank you for the thoughtful question. Our team has published a manuscript reporting outcomes in patients with advanced HCC who received second-line therapy following first-line atezolizumab and bevacizumab therapy. There were no significant differences in survival or time to treatment discontinuation between immune checkpoint inhibitors or anti-VEGF therapies. We have included a brief statement on this study and its conclusions in the Discussion section: ‘Data for 2L therapies in both groups were sparce (Figure 2); most patients received 2L lenvatinib regardless of GI bleeding risk. A previous study by our team found that in patients with advanced HCC, there were no significant differences in survival or time to treatment discontinuation between immune checkpoint inhibitors or anti-vascular endothelial growth factor therapy following 1L atezolizumab + bevacizumab (29).’

Marell P et al. Oncologist 2025;30(8). PMID: 39674576. 

https://pmc.ncbi.nlm.nih.gov/articles/PMC12396946/

(Page #10)

Comments 4: The authors should explain more in details the criteria for unresectability.

Response 4: Thank you for your comment. The criteria for unresectability was based on the surgeon’s assessment, and we have added a statement to clarify this point in the Methods section: ‘Unresectability was based on the surgeon’s assessment ofspecific criteria, including status of infiltrative disease, Barcelona Clinic Liver Cancer (BCLC) C staging, and comorbidities limiting resection.’

(Page #3)

Reviewer 2 Report

Comments and Suggestions for Authors
  1. Given the GI bleeding risk is a key finding associated with the outcome of immunotherapy for uHCC patients, what about the outcome for those uHCC patients with actual GI bleeding, e.g. OS?
  2. Fig. 3B show better OS than Fig. 4B for those without GI bleeding risk, indicating that those 41 patients (58-17) without atezolizumab + bevacizumab could cause the difference; so, what treatments did these patients receive? what about their OS? Additionally, in Fig. 4B, 17 patients are a quite small number.
  3. in the abstract, need to revise "During this study period, many patients with uHCC received atezolizumab + bevacizumab.". This is not a conclusion.

Author Response

Comments 1: Given the GI bleeding risk is a key finding associated with the outcome of immunotherapy for uHCC patients, what about the outcome for those uHCC patients with actual GI bleeding, e.g. OS?

Response 1: As shown in Table 2 on page 6, out of the 128 patients with GI bleeding risk, 25 had a GI bleeding event (19.5%). (Page #6)

We have included a new figure displaying the OS for patients who had a GI bleeding event prior to the index date (n = 41) and a GI bleeding event prior to the index date within 6 months of the pre-index data (n = 25). The median OS for both groups was 12.5 months. (Page #8)

Comments 2: Fig. 3B show better OS than Fig. 4B for those without GI bleeding risk, indicating that those 41 patients (58-17) without atezolizumab + bevacizumab could cause the difference; so, what treatments did these patients receive? what about their OS? Additionally, in Fig. 4B, 17 patients are a quite small number.

Response 2: Please allow us to clarify that the treatment breakdowns for each patient with or without GI bleeding risk are shown in Figure 2 (page 7). Additionally, overall survival (OS) for each group is shown in Figure 4B (page 8). Figure 4B reports the OS for patients who only received atezolizumab + bevacizumab, hence the low number of patients in the no GI bleeding risk cohort. (Page # 7, 8)

We have added the n values for each group in the Discussion: ‘In our exploratory analysis of survival outcomes, median OS with atezolizumab and bevacizumab was 12.8 months in patients with GI bleeding risk (N = 93) and not reached in patients without GI bleeding risk (N = 17), which may be due to the small sample size.’ (Page # 11)

We have also added a statement highlighting the limitation of small sample sizes: ‘It is also relevant to note that the findings of this study relied on the completeness and accuracy of electronic medical record entries which may have contributed to the small sample sizes.’ (Page # 11)

Comments 3: In the abstract, need to revise "During this study period, many patients with uHCC received atezolizumab + bevacizumab.". This is not a conclusion.

Response 3: Thank you for your comment. We have updated the statement in the Abstract conclusions to better describe the finding of the study: ‘During the study period, atezolizumab + bevacizumab was the most common 1L therapy for patients with uHCC, regardless of GI bleeding risk.’ (Page # 1) 

Round 2

Reviewer 1 Report

Comments and Suggestions for Authors

The revised manuscript is OK